**Data Availability Statement:** All relevant data are within the paper and its Supporting information files.

# Evaluation of clinical outcomes of patients with mild symptoms of coronavirus disease 2019 (COVID-19) discharged from the emergency department

Hamidreza Morteza Bagi[1], Maryam Soleimanpour[2], Fariba Abdollahi[3], Hassan Soleimanpour[4]*

1 Emergency Medicine Research Team, Tabriz University of Medical Sciences, Tabriz, Iran, 2 Social Determinants of Health Research Center, Tabriz University of Medical Sciences, Tabriz, Iran, 3 Clinical Skills Center, Faculty of Medicine, Tabriz University of Medical Sciences, Tabriz, Iran, 4 Road Traffic Injury Research Center, Tabriz University of Medical Sciences, Tabriz, Iran

* h.soleimanpour@gmail.com, soleimanpourh@tbzmed.ac.ir

## Abstract

### Introduction

This study was performed to determine the clinical outcomes of patients with mild symptoms of COVID-19 discharged from the emergency department.

### Methods

The present descriptive-analytical cross-sectional study was performed on 400 patients discharged with a diagnosis of COVID-19 from the emergency departments of hospitals affiliated to Tabriz University of Medical Sciences in the time period of 21 March-21 June, 2020. The disease characteristics and demographic data were collected by phone calls during the first, third, and fourth weeks using a researcher-made questionnaire. Finally, the data were analyzed by univariate logistic regression and cross-tabulation using the IBM SPSS Statistics for Windows, version 20.

### Results

In the first week of follow-up, 23(5.8%) patients died, of whom seven patients were female and 16 were male (mean age of death: 70.73±3.27). Out of 41 (10.3%) patients with underlying diseases, 7 (17.1%) died; but out of 359 (89.8%) cases with no history of disease only 16 (4.5%) died. The risk of death in subjects with a history of underlying diseases was 3.27 times higher than those without a history of disease (P = 0.02) (OR = 3.27, 95% CI, 1.20–8.87); and this risk was 1.41 times higher in patients with more family members ($P$ = 0.04) (OR = 1.41, 95% CI, 1.01–1.97). Furthermore, 81 (20.3%) patients had spread the virus to others in their households and disregarded hygiene guidelines such as washing hands, keeping social distancing, and wearing face masks after discharge. In addition, family

**Funding:** The author(s) received no specific funding for this work.

**Competing interests:** The authors have declared that no competing interests exist.

**Abbreviations:** COPD, Chronic obstructive pulmonary disease; ED, Emergency department; OSA, Obstructive sleep apnea.

members of these patients were 16.37 times more likely to be infected than patients who followed the protocols ($P \leq 0.001$) (OR = 16.37, 95% CI, 9.10–29.45).

## Conclusion

Since our findings showed that mortality rate is high in the first week after patients' referral to the emergency departments, the health status of infected people should be carefully monitored daily in this period.

## Introduction

In December 2019, the Chinese government officially announced the outbreak of a novel coronavirus disease in Wuhan, which was then named as COVID-19 [1]. COVID-19 is the seventh coronavirus known to infect humans, and the third zoonotic virus after severe acute respiratory syndrome–related coronavirus (SARS-CoV) and the Middle East respiratory syndrome coronavirus (MERS-CoV), which belong to the beta-coronavirus category [2]. The disease soon spread not only to different cities and regions of China, but also to other Asian countries. Then, the virus spread across Asia and became a global pandemic [3, 4]. On January 30, 2020, the World Health Organization (WHO) declared the pandemic a public health emergency of international concern (PHEIC). Most people experience mild to moderate respiratory illness and are treated without any specific medications. Older people and patients with underlying diseases such as cancer, cardiovascular, and respiratory diseases are more likely to develop severe levels of the disease [3]. The new coronavirus is more widespread in humans than previous coronaviruses, indicating the extremely high transmission rates of the virus [5, 6].

There is a growing concern about the disease for various reasons, including the unknown characteristics of the disease and ways to deal with it, high variety of symptoms, severe infection, long latency period, rapid and high transmission rates, as well as its negative impact on physical and mental health of the communities [7]. Due to the unavailability of antiviral drugs for COVID-19, the WHO recommended the following measurements to reduce the prevalence of infection: practicing personal hygiene and disinfection, early diagnosis and reporting, isolating infected patients, quarantines, proper and rational use of personal protective equipment, social distancing, and travel restrictions [8]. The best way to prevent and control the spread of this disease is to clearly identify the disease and its transmission routes [1]. Although the prevalence of COVID-19 has become a clinical threat to the general population and healthcare staff worldwide, the current knowledge about this new virus is still limited. However, an effective antiviral treatment along with vaccination programs are being evaluated and developed [3]. Currently, aggressive implementation of infection prevention and control (IPC) guidelines is the only option to prevent the spread of human-to-human transmission (HHT) of COVID-19 (3).

Given that COVID-19 pandemic has affected almost all important economic, political, social, and even military systems of all countries worldwide, it is important to discuss the consequences of this viral disease on the health of people at different levels of society [1]. Due to the pathogenesis of the virus and the high spread and mortality rates, COVID-19 can affect the health status of people at different levels of society, including patients, health care staff, families, children, students, and psychiatric patients in different ways. Therefore, in the current high-risk situation, it is necessary to identify the consequences of the disease in order to maintain the health of individuals with appropriate strategies and techniques [8]. Accordingly, this study was carried out to determine the clinical outcomes of patients with mild symptoms of

COVID-19 discharged from the emergency departments of Tabriz University of Medical Sciences, Iran in 2020.

## Materials and methods

The inclusion criteria of this descriptive-analytical cross-sectional study were: patients with mild COVID-19; patients discharged from the emergency department with the aim of home quarantine; and over 18 years of age. By mild COVID-19 symptoms we meant those patients in good general condition, oxygen level ($SpO_2$) over 90%, non-intubated, and having mild fever and also who tested positive for SARS-CoV-2 by reverse transcription polymerase chain reaction (RT-PCR) assay of nasopharyngeal swabs. Patients' unwillingness to participate were excluded from the study. This research was approved by the Research Ethics Committee of Tabriz University of Medical Sciences, Iran with No. IR.TBZMED.REC.1399.630 (14/09/2020). Informed consent was obtained from all participants by telephone at the beginning of the study after explaining the objectives and methods of the study. In addition, participants were assured that their information would remain confidential. The individual in this manuscript has given written informed consent (as outlined in PLOS consent form) to publish these case details.

Data collection tool was a researcher-made questionnaire. The first part of the questionnaire was related to demographic information of patients (age, sex, employment status, marital status, number of family members) and the second part was related to the characteristics of the disease (initial complaint, history of underlying disease, vital signs, the number of referrals to the emergency department, number of infected people in the family, general condition and type of treatment on days 3, 7, and 30, hygiene compliance, reasons for hygiene non-compliance). To determine the reliability of the researcher-made questionnaire, the test-retest method was used on 30 patients; the questions were re-asked by telephone after two weeks, the reliability of which was 0.94. Content validity was used to determine the validity of the questionnaire during the study process. The questionnaire, that had been prepared according to the objectives of the research and review of related literature, was approved by the supervisor. Then, the questionnaire was given to 10 faculty members at Tabriz University of Medical Sciences, and the content validity was approved (S1 and S2 Files). All patients with mild COVID-19 symptoms discharged from the emergency departments of hospitals affiliated to Tabriz University of Medical Sciences in the time period of 21 March-21 June, 2020 were included in the study; the patients were under home quarantine. The sampling method was cluster sampling so that Imam Reza Hospital, Teaching hospital in Tabriz city, with 860 beds and Sina Hospital, Teaching hospital in Tabriz city, with 279 beds as the referral COVID-19 centers, were considered as main clusters. Using patient records, the proportion of patients from each cluster in proportion to the size of that cluster was done randomly. According to Morgan's table, 243 patients from 700 patients referred to the emergency department of Imam Reza Hospital and 152 patients from 300 patients referred to the emergency department of Sina Hospital were selected. The information recorded in the patients' files and the data obtained during the follow-up stage by phone calls were used in the study. Then, all patients were followed up by telephone during the first, third, and fourth weeks. Using the IBM SPSS Statistics for Windows, version 20 (IBM Corp., Armonk, N.Y., USA), data were analyzed using descriptive statistical methods (tables, graphs, mean, and standard deviation) and analytical statistics (univariate logistics and cross-tabulation).

## Results

In this study, out of 400 patients discharged from the emergency departments of Imam Reza and Sina hospitals in Tabriz, 235 (58.8%) were male and 165 (41.3%) were female with a mean

(standard deviation) age of 49.40±17.02 years. Married patients (354 (88.5%)), housewives (122 (30.5%)), pensioners (70 (17.5%)), and freelancers (69 (17.3%)) were the most prevalent groups, respectively. In addition, 130 (32.5%) patients had a family size of four, 108 (27%) had a family size of three, and 63 (15.8%) had a family size of five (Table 1). In addition, 41 (11.2%) patients had a history of underlying diseases, including heart, lung, and kidney diseases (Table 1). The results showed that the mean diastolic blood pressure of the patients was 77.79 ±8.60, the mean systolic blood pressure was 125.33±13.57, the mean heart rate was 85.39 ±13.35, and the mean respiratory rate was 20.40±3.91 (Table 2). In addition, the mean percentage of oxygen saturation in arterial blood was 94.91±2.09. The minimum percentage of oxygen saturation in patients with mild symptoms ranged between 91 and 99. Also, the mean temperature of patients was 37.62 ±0.65 (range: 37 to 39), indicating the presence of fever in patients. Furthermore, the most common and initial complaints of patients were cold symptoms such as fever, myalgia, and cough. Among the participants, 69 (17.3%) patients had referred to the emergency department for two times and 25 (6.3%) patients for three times.

## Medications used in the patients

The results showed that chloroquine was prescribed to most of the patients (49.3%) at the time of discharge from the emergency departments. The frequency of medications prescribed to patients is given in Table 1.

## Chief compliment of patients and so patients need to revisit

In Table 1 chief compliment of patients discharged from ED and so patients need to revisit in the ED has been shown.

## Disease outcomes

In this study, after one week of follow-up, 240 (60%) patients reported their general condition as good and 137 (34.3%) as bad after being discharged from the emergency departments. Unfortunately, 23 (5.8%) patients died in the first week of follow-up, of whom seven were female and 16 were male. Also, the mean age of death was 70.73±3.27 years (Fig 1).

Regarding the association between mortality rate and underlying diseases, the results showed that out of 41 (10.3%) patients with underlying diseases, 7 (17.1%) died; however, there were only 16 (4.5%) death cases out of 359 (89.8%) patients with no history of disease. In the third week, 324 (81%) patients were in good general condition and 53 (13.3%) patients were in bad general condition. Follow-up results on day 30 after discharge showed that 94.3% of patients were in good general condition. Evaluating the relationship between mortality rate with underlying diseases and the number of family members showed that the risk of death in subjects with a history of underlying diseases was 3.27 times higher than those without a history of disease ($P = 0.02$) (OR = 3.27, 95% CI, 1.20–8.87); and this risk was 1.41 times higher in patients with more family members ($P = 0.04$) (OR = 1.41, 95% CI, 1.01–1.97).

## Within-household transmission of COVID-19 and hygiene compliance rate

Regarding the within-household transmission of COVID-19, our results showed that 81 (20.3%) patients had spread the virus to others in their households and disregarded hygiene guidelines such as washing hands, keeping social distancing, and wearing face masks after discharge. Furthermore, two main reasons were identified for non-compliance of hygiene guidelines as follows: small size of the houses (56%) and economic reasons (49%). In addition,

**Table 1. Characteristics of the 400 included patients.**

| variable | | percent | Frequency |
|---|---|---|---|
| **Sex** | Male | 58.8 | 235 |
| | Female | 41.3 | 165 |
| **marital** | Married | 88.5 | 354 |
| | Single | 11.5 | 46 |
| **job** | Retired | 17.5 | 70 |
| | Housewife | 30.5 | 122 |
| | Self-employed | 17.3 | 69 |
| | Unemployed | 2.3 | 9 |
| | Manual worker | 7.2 | 29 |
| | Judge | 0.3 | 1 |
| | Students | 2.5 | 10 |
| | Factory worker | 3.8 | 15 |
| | Employee | 11.8 | 47 |
| | Carpet weaver | 1.5 | 6 |
| | Doctor | 2.3 | 9 |
| | Nurse | 1.8 | 7 |
| | Student | 1.5 | 6 |
| **The Number of family members** | 3 | 27 | 108 |
| | 4 | 32.5 | 130 |
| | 5 | 15.8 | 63 |
| **Past medical history of patients/Underling disease** | NO | 89.9 | 359 |
| | Heart disease | 7.5 | 30 |
| | Kidney disease | 1 | 4 |
| | Pulmonary disease | 0.8 | 3 |
| | Fatty liver | 0.5 | 2 |
| | Compromised immune systems | 0.5 | 2 |
| **Frequency of Prescribed Medications** | No medicine | 12.5 | 50 |
| | Chloroquine, theophylline | 2.0 | 8 |
| | Chloroquine | 49.3 | 197 |
| | Chloroquine, azithromycin | 3.5 | 14 |
| | Azithromycin | 1.5 | 6 |
| | Aminophylline, chloroquine | 11.8 | 47 |
| | Chloroquine, aminophylline and azithromycin | 19.5 | 78 |
| **Chief compliment of patients discharged from ED** | Common cold | 12 | 48 |
| | Weakness and Dizziness | 1.8 | 7 |
| | Chest pain | 4 | 16 |
| | Dyspnea | 6.8 | 27 |
| | Myalgia | 10.3 | 41 |
| | Cough | 12 | 48 |
| | Fever | 11.3 | 45 |
| | Diarrhea and Vomiting | 6.3 | 25 |
| | Sweating | 2.8 | 11 |
| | Headache | 3.8 | 15 |
| | Common cold and Chest pain | 4.3 | 17 |
| | No symptom | 3.5 | 14 |
| | Myalgia and Dizziness | 1 | 4 |
| | Dyspnea and Cough | 2.8 | 11 |
| | Cough and Myalgia | 7 | 28 |
| | Fever and Myalgia | 1.5 | 6 |
| | Fever and Cough | 4.3 | 17 |
| | Vomiting and Fever | 1.3 | 5 |
| | Cough, Anorexia and Myalgia | 2.5 | 10 |
| | Fever, Cough and Myalgia | 1.3 | 5 |
| **Patients need to revisit in the ED** | Second Time | 17.3 | 69 |
| | Third Time | 6.36 | 25 |
| | No | 76.5 | 306 |

**Table 2. Patients' vital signs.**

|       | N   | Minimum | Maximum | Mean     | Std. Deviation |
|-------|-----|---------|---------|----------|----------------|
| SBP   | 400 | 100.00  | 166.00  | 125.3300 | 13.57929       |
| DBP   | 400 | 51.00   | 93.00   | 77.7950  | 8.60891        |
| HR    | 400 | 58.00   | 109.00  | 85.3900  | 13.35398       |
| RR    | 400 | 12.00   | 30.00   | 20.4075  | 3.91468        |
| Sat   | 400 | 91.00   | 99.00   | 94.9150  | 2.09374        |
| GCS   | 400 | 15.00   | 15.00   | 15.0000  | .00000         |
| BT    | 400 | 37.00   | 39.00   | 37.6288  | .65008         |
| Total | 400 |         |         |          |                |

family members of these patients were 16.37 times more likely to be infected than patients who followed the protocols ($P \leq 0.001$) (OR = 16.37, 95% CI, 9.10–29.45).

## Discussion

The COVID-19 virus is mainly transmitted through saliva droplets or nasal discharge when coughing or sneezing [3]. In this study, risk of death in subjects with a history of underlying diseases was higher than others and so 30 (7.5%) of patients had heart disease, 4 (1%) had kidney disease, 3 (0.8%) had lung disease, and 2 (0.5%) had immune system disease. In addition, out of 23 patients who died, seven cases had underlying diseases. Therefore, it can be concluded that patients with underlying diseases such as cancer, cardiovascular, or respiratory disease are more likely to develop severe levels of the disease. Our study showed that 81 (20.3%) patients not only did not follow the health protocols after discharge, but also transmitted the disease to other family members, indicating a significant relationship between the spread of the disease and hygiene non-compliance. In this regard, family members of these patients were 16.37 times more likely to be infected than patients who followed the protocols (P ≤ 0.001) (OR = 16.37, 95% CI, 9.10–29.45). This finding confirms that the COVID-19 is more widespread in humans than previous coronaviruses, indicating its extremely high transmission rate [7]. Studies have shown that the mortality rate of COVID-19 is much lower (4.3%) than other coronaviruses, such as SARS (9.6%) MERS (35.2%), HIV, and Ebola [6, 9–12]. In this study, out of 400 patients, 23 (5.8%) died, which demonstrates a high mortality rate. However, a growing concern has been raised about this disease due to various reasons, including the unknown nature of the disease and ways to deal with it, high variety of symptoms, severe infection, long latency period, and high transmission rate [7]. Studies show that quarantines are used less frequently in large epidemics when people do not support them [7]. Large-scale quarantines can give the health care system a chance to be more resilient to large numbers of patients by delaying the epidemic peak [8]. In our study, the results of examining the relationship between mortality rate and the household size showed that with increasing the number of family members, the risk of death increased by 1.41 times ($P = 0.04$) (OR = 1.41, 95% CI, 1.01–1.97). This indicates the inability to stay in quarantine and hygiene non-compliance for various reasons such as economic reasons, small size of houses, and lack of essential equipment. Wang *et al.* showed that the number of COVID-19 cases in China increased by 31.4 times during 10–24 January, 2020; in addition, on February 23, 2020, the number of infected people in China was 1,879 times higher than January 10, 2020. They estimated the mortality rate as 2.84% based on the number of patients. The researchers also found that the male-to-female mortality rate was 3.25 to 1, the median age of death was 75 years, and the median time from initial symptoms to death was 14 days [13]. In the present study, the mean age of death was

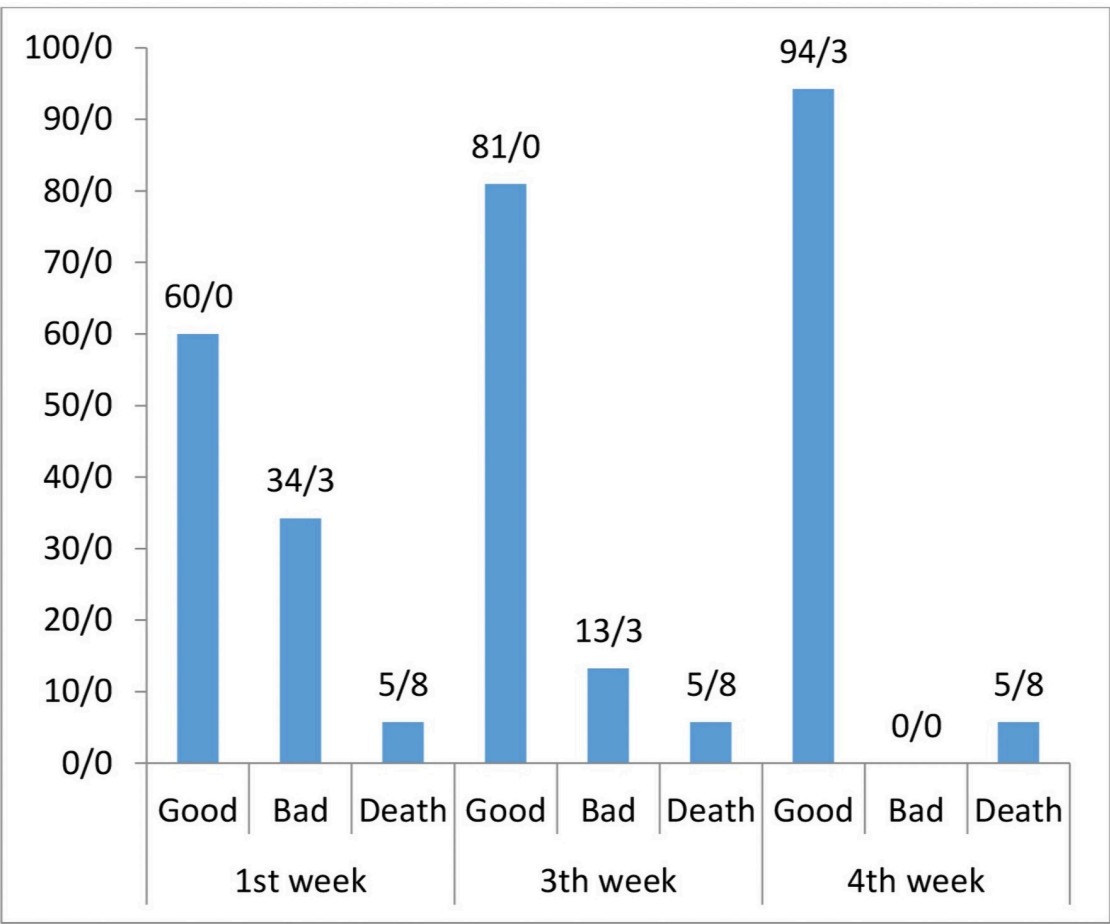

**Fig 1. The frequency of mortality rate.**

70.7 years, and most of them were males who died seven days after referring to the emergency departments. Saad *et al.* reported that the virus could cause severe infections requiring intensive care; concomitant infections and low albumin levels could predict severe infection, whereas age 65 was the only predictor of increased mortality [14]. Liu *et al.* [15] evaluated the effectiveness of hospitalization after two weeks and reported that 11 (14.1%) patients developed severe levels of the disease; meanwhile, 67 (85.9%) patients were treated effectively. Moreover, patients in the severe group were significantly older than patients in the treated group. Several risk factors contributing to the progression of COVID-19 were identified, including age, maximum body temperature at admission, respiratory failure, albumin level, and C-reactive protein (CRP). In this study, hygiene non-compliance, a history of underlying diseases, and a larger family size increased the risk of disease progression. These results can be used in most cases for better management of COVID-19 pandemic. Shahyad *et al.* reported that not only this contagious disease has raised concerns about public physical health, but also it has caused a number of psychological disorders. In these circumstances, maintaining the mental health of communities is essential because people in different social levels may experience stressful stimuli during the pandemic. Therefore, in the current high-risk situation, identifying people prone to psychological disorders at different social levels is necessary to maintain the mental health of individuals with appropriate psychological strategies and

techniques [16]. In the research by Ghonoodi et al., the major barriers for the implementation of quarantine were the lack of knowledge and awareness among communities and financial problems; meanwhile, the main facilitating factor was increasing the knowledge and awareness about quarantine among the communities. For better implementation of quarantines, socio-economic supportive incentives and strengthening of information and educational programs are suggested [17]. In our study, out of 400 patients, 81 individuals did not stay at home due to economic reasons and small house size, which is consistent with the results of Ghonoodi *et* al [17]. Hagh-Ghadam et al. investigated the psychological consequences and interventions during the COVID-19 pandemic in Iran and reported that psychological disorders such as post-traumatic stress disorder (PTSD), depression, anxiety, stress, sleep disorders, and anger were significantly increased in the health care staff and other people involved with COVID-19. Stressors included health anxiety, conspiracy theories, prolonged quarantine, fear of disease transmission, frustration, fatigue, lack of protective equipment, insufficient information, financial loss, rumors, negative beliefs about vaccination, and stigma. In addition, their findings showed that online cognitive-behavioral therapy approach is effective at the time of COVID-19 outbreak. Their review study demonstrated a decrease in mental health of people, especially medical staff. In addition, the researchers recommended that it is necessary to provide fast, continuous, and timely psychological interventions, especially online services. Finally, they suggested that providing online psychological services is better than face-to-face ones [18]. Tavakoli et al. reported the incubation period of COVID-19 as 2–10 days. Overall, the fatality rate of COVID-19 was 4.3%, and the results indicated that the mortality was higher in elderly individuals and patients with chronic conditions, including those with coronary artery disease, diabetes, chronic pulmonary disease, and hypertension. In addition, the mortality rate in healthy subjects was less than 1%, which is consistent with the results of the present study [6]. There was a study in 2020 which had been conducted by Knight et al. It was carried out on 106 patients with mild COVID-19 who were also treated on an out-patient manner. All patients underwent follow up until their PCR tests were negative twice. The average age mean for patients was 51 years and half of them were male. It is about 36.5% of patients had at least one risk factor with asthma (16%) and diabetes (10%) which were the most common types. Most patients (98.1%) had symptoms such as cough (80.4%), fatigue (67.6%), fever (66.0%), headache (49.0%), and ageusia (46.9%). Nine number of patients (8.5%) were admitted in the emergency room and 5 patients (4.7%) were hospitalized. And none of them died. Asthma and immune-compromised were two risk factors with poor prognosis [19]. In our study, number of male patients was 58.8%, 17.5% were retired, 30.5% were housewives, 2.8% were physicians, and 1.8% were nurses. The majority of patients received chloroquine and azithromycin (87.5%) and the rest (12.5%) did not receive any medication. Among 400 discharged patients from the emergency room of Imam Reza general hospital and Sina hospital in Tabriz, 235 numbers were male and 354 were married. The majority of patients had a family of four members. In the first week of patients leave, 240 patients were reported well, 137 patients in bad condition and 23 deaths were reported. After 30 days, 377 patients were reported with good general condition. Results showed that 69 patients (17.25%) referred to the emergency department for the second time and 25 patients (6.25%) for the third time. At all, 23.5% of patients referred to the emergency. After discharge, 81 number of patients were transferred disease to their families. There was a meaningful correlation between disease transmission and the number of family members. The most common symptoms were fever, myalgia and cough. It was about 11.2% of patients had underlying kidney, lung, and heart diseases. In our study, 5.8% of patients (23 numbers) died in the first week with an average age of 70.73 ± 3.27. Findings of the present study showed that 20.3% of patients did not follow hygienic protocols such as hand washing, social distancing and mask usage after discharge. Also, using Chi-square tests, the

results demonstrated that there is a statistically significant relationship between mortality and non-following of health protocol (p = 0.000). In another study conducted by Halalau et al. on 821 patients with an average age of 49.3 years and 46.8% male gender, in 8 hospitals in 2020 year, patients were followed-up at least 12 days after the visit which it included of subsequent ED visit, admission status, and mortality. Cough was the most common symptom of 78.2% with an average of three days long. Other symptoms were 62.1% fever, 35.1% rhinorrhea or nasal congestion and 31.2% dyspnea, respectively. ACEI / ARBs consumption were reported in 28.7% of patients and 34% of patients had a history of diabetes. It was an interesting point that 19.2% of patients were referred to the emergency in which 54.4% of them were admitted. Most of admitted patients were older than discharged patients (mean age 54.4 vs 48.7 years, $p$ = 0.002). Also, hospitalized patients had a higher percentage of blood pressure than those discharged ($p < 0.001$). In addition, the hospitalized patients had a higher BMI than the discharged ones (p = 0.004). Diabetes, pre-diabetes, COPD and OSA increased also in hospitalized in the mentioned individuals rather than out-patients (p < 0.001) [20]. Overall mortality among above mentioned out-patients was 1.3%. Finally, it is hopeful to announce that the majority of patients with COVID-19 recovered without receiving any specific medication, and only sixth of patients developed respiratory problems [21]. And the number of patients who needed ICU was 5% [22]. However, our study was multicenter with relatively modest sample size but larger and further multicenter studies are required to confirm our results.

## Conclusion

Our results confirmed that COVID-19 is more prevalent in males than females, and the disease has a very high transmission rate. There was a significant association between the history of underlying diseases and the higher number of family members with the mortality rate. Moreover, a significant relationship was seen between the number of family members and transmission rate of the disease. Since our findings showed that mortality rate is high in the first week after patients' referral to the emergency departments, the health status of infected people should be carefully monitored daily in this period.

## Supporting information

**S1 File. Original version (Persian) of relevant parts of the questionnaires used in the study.**
(DOC)

**S2 File. Questionnaire.** English translation and so original version (Persian) of relevant parts of the questionnaires used in the study.
(DOC)

## Acknowledgments

This study has been extracted from a M.D dissertation at Tabriz University of Medical Sciences, Iran. The authors wish to thank the Clinical Research Development Unit, Imam Reza General Hospital, Tabriz University of Medical Sciences, Tabriz, Iran and so the Clinical Research Development Unit of Sina Educational, Research and Treatment Center, Tabriz University of Medical Sciences, Tabriz, Iran for their sincere collaboration.

## Author Contributions

**Conceptualization:** Hamidreza Morteza Bagi.

**Investigation:** Hamidreza Morteza Bagi, Fariba Abdollahi.

**Methodology:** Hamidreza Morteza Bagi, Maryam Soleimanpour.

**Supervision:** Hassan Soleimanpour.

**Writing – original draft:** Hassan Soleimanpour.

**Writing – review & editing:** Maryam Soleimanpour, Hassan Soleimanpour.

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
