## [Decision Letter · Decision Letter 0]

23 Aug 2021

PONE-D-21-20101

Evaluation of Clinical Outcomes of Patients with Mild Symptoms of Coronavirus Disease 2019 (COVID-19) Discharged from the Emergency Departments

PLOS ONE

Dear Dr. Soleimanpour,

Thank you for submitting your manuscript to PLOS ONE. After careful consideration, we feel that it has merit but does not fully meet PLOS ONE’s publication criteria as it currently stands. Therefore, we invite you to submit a revised version of the manuscript that addresses the points raised during the review process.

We look forward to receiving your revised manuscript.

Kind regards,

Edris Hasanpoor

Academic Editor

PLOS ONE

Journal Requirements:

2. We note that your paper includes detailed descriptions of individual patients/participants. As per the PLOS ONE policy (http://journals.plos.org/plosone/s/submission-guidelines#loc-human-subjects-research) on papers that include identifying, or potentially identifying, information, the individual(s) or parent(s)/guardian(s) must be informed of the terms of the PLOS open-access (CC-BY) license and provide specific permission for publication of these details under the terms of this license. Please download the Consent Form for Publication in a PLOS Journal (http://journals.plos.org/plosone/s/file?id=8ce6/plos-consent-form-english.pdf). The signed consent form should not be submitted with the manuscript, but should be securely filed in the individual's case notes. Please amend the methods section and ethics statement of the manuscript to explicitly state that the patient/participant has provided consent for publication: “The individual in this manuscript has given written informed consent (as outlined in PLOS consent form) to publish these case details

5. Please ensure that you refer to Figure 1 in your text as, if accepted, production will need this reference to link the reader to the figure.

6. We note you have included a table to which you do not refer in the text of your manuscript. Please ensure that you refer to Table 1 in your text; if accepted, production will need this reference to link the reader to the Table.

Reviewers' comments:

Reviewer's Responses to Questions

**Comments to the Author**

1. Is the manuscript technically sound, and do the data support the conclusions?

Reviewer #1: Yes

Reviewer #2: Partly

2. Has the statistical analysis been performed appropriately and rigorously? 

Reviewer #1: Yes

Reviewer #2: Yes

3. Have the authors made all data underlying the findings in their manuscript fully available?

Reviewer #1: Yes

Reviewer #2: Yes

4. Is the manuscript presented in an intelligible fashion and written in standard English?

Reviewer #1: Yes

Reviewer #2: Yes

5. Review Comments to the Author

Reviewer #1: Dear Associate editor

Thanks to the authors. I reviewed the article. It is valuable research in the pandemic era. The title, introduction, methodology, and conclusion are appropriate. There is no plagiarism in it. The writing is appropriate and completely understandable. Moreover, recommendations are consistent with the finding, and referencing and citations were correct and so English writing was acceptable. The article needs minor revision and can be acceptable for publication. My comments are as follows:

1- The authors do not discuss the specifics of the hospital involved in the study.

2- Please mention the weak and strong points of your study

Reviewer #2: 1-Is there any information regarding the severity scores like APACHE or SAPS in the study population?

2-There should be a table showing demographic characteristics of the patients in the study.

3-is there any information about the duration of ED stay, method of ventilation, prescribed drugs, antiviral drugs, ...?

4-What was the criteria used for mild category of COVID-19 in this study?

5-What was the criteria for discharge of patients from ED?

6-it is better to show the any information about ED revisits, inpatient admissions, ICU admissions after discharge from hopital.

7-How did the authors delete the effect of confounding factors from the study outcome?

8-Discussion should be improved and revised based on the study results and comparison with previous ones.

9-What is the new finding of this study compared to previous ones the this field?

10-Is there any data regarding economic situation of patients,as previous studies showed an important economic consequence of the disease at an early post-hospital phase.

11-What was the limitation of the study?

6. PLOS authors have the option to publish the peer review history of their article (what does this mean?). If published, this will include your full peer review and any attached files.

Reviewer #1: No

Reviewer #2: **Yes: **Ata Mahmoodpoor

---

## [Author Response · Author response to Decision Letter 0]

13 Sep 2021

Dear Respected Editor and reviewers

Many thanks for your comprehensive comments regarding our article. Herewith, we revised our article based on your advice in the best manner as follows:

Reviewer #1: 

Dear Reviewer 1

I appreciate you for reviewing our manuscript so meticulously and mentioning very important comments. Below you could find our reply to your comments.

Q1: The authors do not discuss the specifics of the hospital involved in the study.

Reply: Based on your comment, we explained the specifics of the hospitals involved in the study in the Method section.

 Q2: Please mention the weak and strong points of your study

Reply: Based on your comment, we explained the weak and strength of study in end of discussion section.

Reviewer #2:

Dear Reviewer 2

I appreciate you for reviewing our manuscript so meticulously and mentioning very important comments. Below you could find our reply to your comments.

Q 1-Is there any information regarding the severity scores like APACHE or SAPS in the study population?

Reply: Thank you so much for reviewing our manuscript meticulously. Since our included patients were outpatient’s patients we couldn’t collected data for calculating the APACHE or SAPS due to cost problem. Furthermore, based on the literature and experts’ opinion the mentioned predictor scores is not suitable for COVID-19 patients. 

Q2- There should be a table showing demographic characteristics of the patients in the study.

Reply: I really appreciate you for your comprehensive comments. We provided 3 tables (1, 2 and 5) regarding your nice advice.

Q3-is there any information about the duration of ED stay, method of ventilation, prescribed drugs, antiviral drugs...?

Reply: We studied discharged outpatient patients from ED no need to ventilation or ED stay. The prescribed drug is mentioned in Table 4.

Q4-What was the criteria used for mild category of COVID-19 in this study?

Reply: We meant those patients in good general condition, oxygen level (SpO2) over 90%, non-intubated, and having mild fever and also who tested positive for SARS-CoV-2 by reverse transcription polymerase chain reaction (RT-PCR) assay of nasopharyngeal swabs.

Q5-What was the criteria for discharge of patients from ED?

Reply: We discharged patients in good general condition, oxygen level (SpO2) over 90%, non-intubated, and having mild fever and also who tested positive for SARS-CoV-2 by reverse transcription polymerase chain reaction (RT-PCR) assay of nasopharyngeal swabs.

6-it is better to show the any information about ED revisits, inpatient admissions, ICU admissions after discharge from hospital.

Reply: Thank you for your valuable comment. The required data has been show in the Table 6 and figure 1

7-How did the authors delete the effect of confounding factors from the study outcome?

Reply- : We meant those patients in good general condition, oxygen level (SpO2) over 90%, non-intubated, and having mild fever and also who tested positive for SARS-CoV-2 by reverse transcription polymerase chain reaction (RT-PCR) assay of nasopharyngeal swabs.

8-Discussion should be improved and revised based on the study results and comparison with previous ones.

Reply- Many thanks for your attention. We did comprehensive review and improved the discussion section by the similar studies.

Q9-What is the new finding of this study compared to previous ones the this field?

Reply- We discussed the mortality, Household Transmission of COVID-19 and Hygiene Compliance Rate and so revisit of the Mild covid-19 outpatients discharged from ED. 

10-Is there any data regarding economic situation of patients,as previous studies showed an important economic consequence of the disease at an early post-hospital phase.

Reply- In Table 1 the patients’ job is explained. 

11-What was the limitation of the study?

Reply- Based on your comment, we explained the weak and strength of study in end of discussion section.

---

## [Decision Letter · Decision Letter 1]

4 Oct 2021

Evaluation of Clinical Outcomes of Patients with Mild Symptoms of Coronavirus Disease 2019 (COVID-19) Discharged from the Emergency Departments

PONE-D-21-20101R1

Dear Dr. Soleimanpour

We’re pleased to inform you that your manuscript has been judged scientifically suitable for publication and will be formally accepted for publication once it meets all outstanding technical requirements. Within one week, you’ll receive an e-mail detailing the required amendments. When these have been addressed, you’ll receive a formal acceptance letter and your manuscript will be scheduled for publication.

Kind regards,

Edris Hasanpoor

Academic Editor

PLOS ONE

---

## [Editor Report · Acceptance letter]

13 Oct 2021

PONE-D-21-20101R1 

Evaluation of Clinical Outcomes of Patients with Mild Symptoms of Coronavirus Disease 2019 (COVID-19) Discharged from the Emergency Department 

Dear Dr. Soleimanpour:

I'm pleased to inform you that your manuscript has been deemed suitable for publication in PLOS ONE. Congratulations! Your manuscript is now with our production department. 

Kind regards, 

on behalf of

Dr. Edris Hasanpoor 

Academic Editor

PLOS ONE